# Dairy Consumption and 3-Year Risk of Type 2 Diabetes after Myocardial Infarction: A Prospective Analysis in the Alpha Omega Cohort

**DOI:** 10.3390/nu13093146

**Published:** 2021-09-09

**Authors:** Maria G. Jacobo Cejudo, Esther Cruijsen, Christiane Heuser, Sabita S. Soedamah-Muthu, Trudy Voortman, Johanna M. Geleijnse

**Affiliations:** 1Division of Human Nutrition and Health, Wageningen University, P.O. Box 17, 6700 AA Wageningen, The Netherlands; esther.cruijsen@wur.nl (E.C.); christiane.heuser@gmx.com (C.H.); trudy.voortman@wur.nl (T.V.); marianne.geleijnse@wur.nl (J.M.G.); 2Center of Research on Psychological and Somatic Disorders (CORPS), Department of Medical and Clinical Psychology, Tilburg University, P.O. Box 90153, 5000 LE Tilburg, The Netherlands; S.S.Soedamah@tilburguniversity.edu; 3Institute for Food, Nutrition and Health, University of Reading, Reading RG6 6AR, UK; 4Department of Epidemiology, Erasmus MC, University Medical Center Rotterdam, P.O. Box 2040, 3000 CA Rotterdam, The Netherlands

**Keywords:** dairy, yogurt, type 2 diabetes incidence, myocardial infarction, prospective cohort study

## Abstract

Population-based studies suggest a role for dairy, especially yogurt, in the prevention of type 2 diabetes (T2D). Whether dairy affects T2D risk after myocardial infarction (MI) is unknown. We examined associations of (types of) dairy with T2D incidence in drug-treated, post-MI patients from the Alpha Omega Cohort. The analysis included 3401 patients (80% men) aged 60–80 y who were free of T2D at baseline (2002–2006). Dairy intakes were assessed using a validated food-frequency questionnaire. Incident T2D was ascertained through self-reported physician diagnosis and/or medication use. Multivariable Cox models were used to calculate Hazard ratios (HRs) and 95% confidence intervals (CI) for T2D with dairy intake in categories and per 1-standard deviation (SD) increment. Most patients consumed dairy, and median intakes were 264 g/d for total dairy, 82 g/d for milk and 41 g/d for yogurt. During 40 months of follow-up (10,714 person-years), 186 patients developed T2D. After adjustment for confounders, including diet, HRs per 1-SD were 1.06 (95% CI 0.91–1.22) for total dairy, 1.02 (0.88–1.18) for milk and 1.04 (0.90–1.20) for yogurt. Associations were also absent for other dairy types and in dairy categories (all *p*-trend > 0.05). Our findings suggest no major role for dairy consumption in T2D prevention after MI.

## 1. Introduction

Type 2 diabetes (T2D) is a major public health problem, especially in aging populations [1,2], with a global prevalence in the elderly of approximately 20% (135.6 million) in 2019 [1]. T2D and coronary heart disease (CHD) are closely interrelated. Patients who suffered a myocardial infarction (MI) often experience impaired glucose metabolism, which may evolve into T2D [3].

Diet plays a crucial role in the development of T2D [4,5,6]. In 2017, suboptimal diet was responsible for 25% of attributable deaths and 35% of disability-adjusted life years to diabetes globally [7,8]. Thus, the identification of modifiable dietary factors for the prevention of T2D is of public health importance, not only in the general population but also specifically in patients with CHD. Milk and dairy products are major components of diets in northwestern Europe and North America [9]. Average dairy intake in the Netherlands is ~390 g/d for men and 325 g/d for women [10], and in the US is on average 1.5 cup-servings/d which equals around ~356 g/d of milk or equivalent portions of yogurt, cheese and other dairy products [9,11].

Dairy products constitute a heterogeneous group of foods with complex matrices of nutrients that include calcium, magnesium, potassium, vitamin D, proteins, different types of fatty acids, among others including added sodium, that can exert differential health effects [12,13]. Dairy consumption could play a role in the prevention of T2D, which may depend on the type of dairy [14,15,16]. Previous meta-analyses and systematic reviews of prospective cohort studies in generally healthy populations have shown that consumption of total dairy, low-fat dairy and especially yogurt, has been neutrally or weakly inversely associated with T2D incidence [15,16,17]. Results for cheese and T2D risk differ depending on the studies included in the meta-analyses [14,15,16,17,18,19], while milk and other dairy products are generally neutrally associated with T2D risk.

In Dutch post-MI patients from the Alpha Omega Cohort, a healthy dietary pattern was associated with a ~30% lower mortality risk [20], on top of cardiovascular drug treatment. In a more recent analysis in this cohort, yogurt consumption was inversely associated with cardiovascular mortality (HR: 0.96; 95% CI: 0.93 to 0.99, per 25 g/d) [21]. Whether dairy consumption affects the risk of T2D after MI is at present unknown. Therefore, we examined total dairy intake and a wide range of dairy products in relation to incident T2D during 40 months of follow-up in our cohort of post-MI patients of the Alpha Omega Cohort.

## 2. Materials and Methods

The Alpha Omega Cohort (AOC) emerged from the Alpha Omega Trial, a 3-y (40 months) intervention study of low doses of omega 3 fatty acids which did not affect major cardiovascular disease (CVD) events, as described in detail elsewhere [22,23]. The cohort is registered at clinicaltrials.gov (ID NCT03192410; link: https://clinicaltrials.gov/ct2/show/NCT03192410 accessed on 20 July 2021).

### 2.1. Study Design and Population

The AOC is a prospective cohort study of 4837 drug-treated Dutch patients between 60–80 y at baseline (2002–2006), with a history of MI ≤ 10 y before study enrollment [22,23]. Patients were recruited from 32 hospitals in the Netherlands. The study was approved by a central ethics committee at the Haga Hospital (Leyenburg, The Hague, The Netherlands), and by the ethics committees at each participating hospital. All patients provided written informed consent.

For the present study, we excluded patients with incomplete dietary data (*n* = 453), implausible reported energy intakes (<600 or >6000 kcal/d for women and <800 or >8000 kcal/d for men, *n* = 19), and prevalent diabetes (*n* = 883) at baseline. We further excluded patients for which T2D incidence could not be assessed (no blood samples and no follow-up data on medication: *n* = 22), and patients who died during early follow-up (*n*= 59), of which 44% from CVD. Mortality risk during 3 year follow-up did not differ across categories of total dairy intake (1.8% for highest vs. 1.6% for lowest category). After exclusions, the analysis included 3401 patients (Appendix A).

### 2.2. Dietary Assessment

Self-reported dietary intake was assessed at baseline using a semi-quantitative 203-item food-frequency questionnaire (FFQ), which was an adapted and extended version of a previously validated and reproducible FFQ, specifically designed to estimate fatty acids and cholesterol intake [24,25]. The Pearson’s correlation coefficient for estimated energy intake was 0.83 for the FFQ compared to a dietary history measured over the same period [24], and Spearman’s correlation coefficients were 0.69 for cheese and 0.80 for milk, yogurt and custard combined, indicating high validity and reproducibility [25]. Patients were asked to report their habitual food intake in the previous month, including type of food, frequency, amount and preparation method if applicable. Dairy intake was covered by 42 items grouped by fat content. Intake of butter, milk and creamers from non-dairy sources, such as soy milk, were not included in this study. Dairy intake was estimated in g/d, and grouped by dairy type as follows: total dairy, total milk, low-fat milk, full-fat milk, hard cheese, total (plain) yogurt, low-fat yogurt, full-fat yogurt, total fermented dairy, liquid fermented dairy, (ice)-cream, and dairy desserts (Appendix A), as reported previously [21]. Daily intake of total energy (kcal/d) and nutrients (g/d) was calculated through linkage with the 2006 Dutch Food Composition Database (NEVO 2006) [26]. The year 2006 was chosen because this corresponded with the baseline dietary assessments (2002–2006).

### 2.3. Ascertainment of Incident Type 2 Diabetes

Diabetes at baseline was defined based on WHO guidelines [27]: plasma glucose levels ≥ 7.0 mmol/L (126 mg/dL) in fasting conditions (≥4 h), or ≥11.1 mmol/L (200 mg/dL) in non-fasting conditions, or having received the diagnosis from a physician (based on WHO guidelines), and/or taking antidiabetic drugs. New cases of T2D were ascertained based on a self-reported physician’s diagnosis and/or start of antidiabetic medication use. T2D was assessed by telephone interviews at 12 and 24 months, or during examinations at 20 months (in a subsample of 810 participants), or 40 months, with a median (interquartile range: IQR) follow-up of 40 (37.0–41.0) months. For the cases in which the date of diagnosis or start of medication was not available, it was calculated as the midpoint between two interview dates (10% of cases) [28]. Since incident T2D during early follow-up (i.e., initial trial phase) was the main outcome for this study, prevalent cases of diabetes at baseline were excluded.

### 2.4. Data Collection on Risk Factors/Covariates

At baseline, data were collected on demographic factors, lifestyle, medical history and current health status through questionnaires as previously described [23]. Educational level was assessed in three categories (low, intermediate and high education). Smoking status was assessed in three categories (never, former or current). Physical activity was assessed with the Physical Activity Scale for the Elderly [29], and patients were categorized according to the number of days per week of at least 30 min of moderate and/or vigorous leisure activity (>3 metabolic equivalents (METs)) as follows: low (≤3 METs), intermediate (>3 METs) 1 to <5 days/week, and high (>3 METs) ≥5 days/week. Alcohol consumption was estimated from the FFQ in g/d and was categorized into: abstainers, light (>0–≤10 for males or >0–≤5 for females), moderate (>10–≤30 for males or >5–≤15 for females) and heavy (>30 for males or >15 for females). Body weight (kg) and height (cm) were measured with the subject wearing light indoor clothing without shoes, and the body mass index (BMI, kg/m^2^) was computed. Obesity was defined as BMI ≥ 30 kg/m2. Waist circumference (cm) was measured at the midpoint between the bottom rib and the top of the hipbone.

Blood pressure was measured twice using an automated device (Omron HEM-711), and the average of the two measurements was reported. Medication use was self-reported and coded according to the Anatomical Therapeutic Chemical Classification System (ATC) [30]. ATC codes were C02, C03, C07, C08 and C09 for antihypertensive drugs, C10 for lipid-modifying drugs including statins (C10AA), and A10 for antidiabetic drugs. Blood lipids and glucose levels were measured in an autoanalyzer Hitachi 912 (Roche Diagnostics, Basel, Switzerland) using standard kits. Family history of T2D was considered present when patients reported at least one parent having T2D. Hypertension was considered as use of antihypertensive drugs or either systolic blood pressure ≥ 140 mm/Hg or diastolic blood pressure ≥ 90 mm/Hg. Hypercholesterolemia was considered as use of lipid-lowering drugs or total serum cholesterol levels ≥ 5 mmol/L.

### 2.5. Statistical Analysis

Missing data (all assumed to be at random) were imputed using age and sex specific medians for BMI (*n* = 4), and using age and sex specific mode for the categorical covariates physical activity (*n* = 17), smoking status (*n* = 1), family history of diabetes (*n* = 11), and educational level (*n* = 15). Dairy intakes were adjusted for total energy intake with the residual method [31], and standardized to the study population’s mean energy intake of 1946 kcal/d. Baseline characteristics of patients are presented for the total population and across categories of energy-adjusted total milk intake (the most consumed type of dairy), as mean ± SD for normally distributed data, median (IQR) for skewed data and % (*n*) for categorical data.

Cox proportional hazards regression models were used to obtain Hazard Ratios (HRs) with 95% confidence intervals (95% CIs) for incident T2D across categories of types of dairy, using the lowest category of consumption as the reference. The following categories of dairy (types) intake in g/d were used: total dairy (<200, ≥200–<400, ≥400), total milk (<50, ≥50–<150, and ≥150), low-fat milk (<50, ≥50–<150, and ≥150), hard cheese (<15, ≥15–<30, and ≥30), total yogurt (<25, ≥25–<50, and ≥50), low-fat yogurt (<25, ≥25–<50, and ≥50), total fermented dairy (<50, ≥50–<100, and ≥100) and dairy desserts (<30, ≥30–<60, and ≥60). Full-fat milk, full-fat yogurt, liquid fermented dairy and (ice)-cream, for which intake was relatively low, were assessed dichotomously (any intake vs. no intake) without energy-adjustment. Hazard ratios for T2D per 1-standard deviation (SD) increment in types of dairy consumption (g/d), were also calculated using energy-adjusted dairy intakes.

The proportional hazards assumption was confirmed visually using a log-minus-log plot of the survival time. Person-years were calculated from time of study enrolment to date of T2D incidence, death, lost to follow-up or end of the follow-up period (through November 2009), whichever came first.

Multivariable models included potential confounders and established risk factors for T2D. Model 1 was adjusted for age (y), sex and total energy intake (kcal/d). Model 2 was additionally adjusted for smoking (three categories), physical activity (three categories), educational level (three categories), alcohol intake (four categories), family history of diabetes (yes/no) and BMI (kg/m^2^). Model 3 was additionally adjusted for the following dietary intakes in g/d: whole grains, refined grains, potatoes, fruit, vegetables, total red and processed meat, sugar-sweetened beverages, caffeinated and decaffeinated coffee and green and black tea. Model 4 was additionally adjusted for all other types of dairy (except for analysis of total dairy). In exploratory analyses, BMI was excluded from multivariable analyses (models 2 through 4) because of its potential intermediary role in the dairy-diabetes pathway.

We tested for linear trends across categories of types of dairy consumption by treating the median value of each category as a continuous variable in the models. Non-linear associations were tested by analyzing types of dairy continuously with the use of natural cubic splines (df = 3) in crude models, and as no indications for non-linear associations were found (*p*-values ≥ 0.05), all analyses were performed assuming linearity.

In ancillary analyses, dairy consumption was studied in relation to three year changes in body weight (cm), BMI and waist circumference (cm), as major risk factors for T2D, in a subsample of 2313 surviving patients (first part of the cohort, enrolled before August 2005) who were re-examined by a research nurse at the end of follow-up (Appendix A). We used general linear models to calculate β coefficients and 95% CI for anthropometric outcomes in categories of types of dairy, using the lowest category of consumption or no consumption as the reference. Multivariable models were adjusted for baseline body weight, BMI or waist circumference, age, sex, total energy intake, smoking, physical activity, educational level, alcohol intake, whole grains, refined grains, potatoes, fruit, vegetables, legumes, total red and processed meat, fish, sugar-sweetened beverages, coffee, tea, and other types of dairy (except for total dairy).

We performed subgroup analyses by obesity status at baseline and by sex for the associations of total dairy, total milk, low-fat milk, hard cheese, total yogurt, low-fat yogurt, total fermented dairy and dairy desserts with T2D risk, using model 4.

SPSS Statistics version 25.0 (IBM SPSS Inc., Chicago, IL, USA) was used for data-analysis, and non-linear associations were additionally examined in R (version 3.6.1; R Foundation for Statistical Computing). Two-sided *p*-values < 0.05 were considered as statistically significant.

## 3. Results

Patient characteristics are presented in Table 1. The mean age was 68.9 ± 5.5 years, most patients were men (80%), and 20% had obesity. Most patients had hypercholesterolemia (96%) and hypertension (95%), for which 87% used statins and 89% antihypertensive drugs. During a median follow-up time of 40 months (10,714 person-years), 186 new cases of T2D were diagnosed. No patients were lost during follow-up.

Median dairy intakes (without energy-adjustment) are presented in Appendix A, and were 264 g/d for total dairy, 82 g/d for total milk, 64 g/d for low-fat milk, 16 g/d for hard cheese, 41 g/d for total yogurt, 32 g/d for low-fat yogurt, 154 g/d for total fermented dairy and 43 g/d for dairy desserts. Full-fat milk was consumed by 10% of patients (median intake of 53 g/d in consumers), full-fat yogurt by 30% (median intake of 15 g/d in consumers), liquid fermented dairy by 38% (75 g/d in consumers) and (ice)-cream by 66% (13 g/d in consumers).

### 3.1. Dairy Consumption and T2D Incidence

Associations for main types of dairy and T2D risk are presented in Table 2, and associations for full-fat milk, full-fat yogurt, liquid fermented dairy and (ice)-cream are presented in Appendix A. After adjustment for demographics, lifestyle, BMI, family history of diabetes and dietary variables (model 3), total dairy was not associated with T2D risk (HR: 1.32, 95% CI: 0.88 to 1.99, highest vs. lowest category, *p*-trend = 0.18). Also, no associations were found for total milk, low-fat milk, full-fat milk, hard cheese, total yogurt, low-fat yogurt, full-fat yogurt, liquid fermented dairy, total fermented dairy, (ice)-cream or dairy desserts (highest vs. lowest category or any intake vs. no intake), and T2D incidence (model 4), with HRs ranging from 0.80 (95% CI: 0.47 to 1.38, *p*-value = 0.43) for full-fat milk to 1.19 (95% CI: 0.86 to 1.64, *p*-trend = 0.24) for low-fat yogurt. Associations of types of dairy with T2D risk were also absent when dairy intakes were not adjusted for total energy intake (results not shown).

Figure 1 shows HRs for T2D per 1-SD increment in types of dairy consumption, showing no significant associations in line with results for dairy in categories (Table 2) or as dichotomous variables (Appendix A).

### 3.2. Exploratory Analyses

Results of analyses on the potential intermediary role of BMI between dairy and T2D are presented in Appendix A for total dairy, total milk, low-fat milk, hard cheese, total yogurt, low-fat yogurt, total fermented dairy and dairy desserts, and in Appendix A for full-fat milk, full-fat yogurt, liquid fermented dairy and (ice)-cream. Exclusion of BMI from models 2 through 4 yielded essentially similar results for types of dairy consumption and T2D risk.

### 3.3. Ancillary Analyses

After three year follow-up, no significant overall changes were observed in body weight, BMI and waist circumference in 2313 patients from the AOC (Appendix A). After multivariable adjustment, total dairy (highest vs. lowest category) was not associated with changes in body weight (β: −0.16 kg, 95% CI: −0.67 to 0.36, *p*-value = 0.55), BMI (β: −0.06 kg/m2, 95% CI: −0.25 to 0.13, *p*-value = 0.55), or waist circumference (β: 0.04 cm, 95% CI: −0.64 to 0.72, *p* = 0.90), after 3-y of follow-up. Other types of dairy were also not associated with changes in anthropometric outcomes during follow-up (Appendix A).

### 3.4. Subgroup Analyses

Results of subgroup analyses are presented in Appendix A. When examining associations between dairy consumption and T2D risk by obesity status at baseline, in patients with obesity, dairy desserts (highest vs. lowest category) were associated with higher T2D risk (HR: 2.58, 95% CI: 1.17 to 5.69, *p*-trend = 0.041), and a trend toward higher T2D risk was observed with higher low-fat yogurt consumption (*p*-trend = 0.041), whereas all other types of dairy were not associated with incident T2D (Appendix A). No associations were observed for any of the types of dairy analyzed and T2D risk in patients without obesity (Appendix A).

When examining associations between dairy consumption and T2D risk by sex, we observed a trend toward higher T2D risk in women for higher total dairy and total milk consumption (*p*-trends = 0.016 and 0.023 respectively, Appendix A). In men, no associations were observed for any of the types of dairy analyzed and T2D risk (Appendix A).

## 4. Discussion

In this prospective cohort study of Dutch post-MI patients from the Alpha Omega Cohort, consumption of total dairy or types of dairy was not associated with three year risk of T2D.

To the best of our knowledge, no other prospective cohort studies of dairy and incident T2D among patients with stable CHD are available for comparison. In the general population, previous meta-analyses of prospective population-based cohort studies have shown neutral or modest linear inverse associations for total dairy and low-fat dairy consumption with T2D risk. Yogurt consumption has been non-linearly inversely associated with T2D incidence in several meta-analysis [15,16,17]. Results for cheese differ between meta-analyses depending on the studies included, showing both inverse and neutral associations with T2D incidence [14,15,16,17,18,19], while milk and other dairy products tend not to be associated with T2D risk.

Dairy products are a heterogeneous group of foods with a complex matrix of nutrients that may interact with other bioactive compounds, food structures or probiotics, resulting in complex cardiometabolic health effects [12]. Dairy consumption may influence various metabolic pathways linked to cardiometabolic health and disease, including insulin resistance and diabetes, and these effects appear to be dairy type specific. Recent scientific evidence suggests that fermented dairy, particularly yogurt, could exert anti-diabetic properties through effects on satiety and risk of adiposity, and/or directly on insulin sensitivity [32]. Potential antidiabetic effects of yogurt may be driven by nutrients like calcium, protein, bioactive peptides and/or probiotics, and the interactions between them [33].

We did not find associations between any of the type of dairy we studied and T2D in our post-MI patients. Milk consumption was the most consumed dairy food in the present cohort, contributing 40% to total dairy intake. Our results for total (non-fermented) milk are in line with results from previous meta-analyses of observational population-based cohort studies, showing no associations for milk consumption and T2D risk [14,15]. Our results for yogurt are in contrast with previously reported beneficial associations for yogurt and T2D incidence in population-based studies [15,16,17]. We studied a cohort of drug-treated MI survivors, who may respond differently to dairy intake compared to healthy populations. Another possible explanation for this difference could be the relatively short follow-up of our cohort, since differences in associations between types of dairy consumption and T2D may differ by follow-up time [15,16,17]. Also, results from meta-analyses need to be interpreted with caution, since not all individual prospective cohort studies showed an inverse association between yogurt consumption and T2D risk, and results from meta-analyses were found in the context of significant heterogeneity. Furthermore, to date, the potential benefit of yogurt against T2D is largely supported by observational evidence [33,34,35,36].

Dairy consumption may influence markers of adiposity like body weight and thus BMI [37,38], and a higher BMI has been associated to greater risk of T2D [39]. BMI may thus have a mediating role in the associations between dairy consumption and T2D. However, we found no significant associations of types of dairy consumption with 3-y changes in body weight, BMI and waist circumference. We also did not observe differences in the estimates when excluding adjustment for BMI in our models for dairy and T2D.

A previous systematic review that included 10 observational prospective cohort studies, suggested that yogurt consumption may be inversely associated with changes in body weight and waist circumference over time in the general population [40], and an observational study with three American cohorts showed that greater yogurt consumption, but no other types of dairy, was weakly inversely associated with less weight gain after 24 y of follow-up in a general population (−41 g/y per daily serving of yogurt) [37]. However, the findings from a meta-analysis of 27 RCTs that included 2101 participants from general populations suggested that increased dairy consumption (combination of milk, cheese and yogurt) was associated with lower body weight but only during energy restriction, with opposite or null effects in ad libitum conditions, and only during short follow-up (<1 y), not during follow-up >1 y [41].

Results from subgroup analysis showed that higher consumption of dairy-based desserts was associated with an increased risk of T2D in 685 post-MI patients with obesity. Dairy desserts with added sugar contributed substantially to total dairy intake (19%), which may partly explain why we found no inverse associations between total dairy and T2D. In the subgroup of 680 women, we observed a trend towards a higher risk of T2D with total dairy and total milk intake. However, findings in subgroups need to be interpreted with caution because of small sample sizes, as reflected also by wide confidence intervals.

Dietary guidelines for adult populations generally include two–three daily servings of dairy [10,42]. In our cohort of post-MI patients, we previously found an inverse association between (plain) yogurt consumption and long-term risk of CVD mortality [21]. The present analysis showed neutral findings for T2D, so the cardioprotective association that we observed for yogurt may be related to other (yet unknown) pathways. Based on the totality of findings in our Alpha Omega Cohort, we conclude that dairy products could fit in a diet for CHD patients, if consumed in moderation and without added sugar.

Strengths of this study are its prospective design, and the well-documented population of post-MI patients in AOC with extensive data collection on potential confounders. Also, we used a detailed, validated FFQ with high reproducibility for milk, yogurt and cheese with different fat content [25]. This study also had limitations. The number of incident T2D cases was relatively low, and follow-up time was relatively short. Most patients in our study were Dutch men. Therefore, results cannot merely be translated to people of non-Caucasian origin and to women. Yet, although dairy consumption and T2D risk may vary by sex, we found no significant interaction terms between dairy and sex in our statistical analysis. Also, we have no biological explanation why men would respond differently to dairy intake than women. Another limitation is the small number of patients who consumed no dairy, which hampered the analysis of very low dairy intake, and finally we cannot completely rule out misclassification of patients due to lack of repeated assessment of dietary intake, even when previous studies in Dutch elderly have shown reasonably stable dietary patterns over time [43], which could have led to attenuated risk estimates.

## 5. Conclusions

To conclude, intake of total dairy and different types of dairy was neutrally associated with T2D incidence in drug-treated post-MI patients from the Netherlands. Our findings warrant further confirmation in other cohorts of CHD patients with a larger number of women and different ethnic backgrounds.

## Figures and Tables

**Figure 1 nutrients-13-03146-f001:**
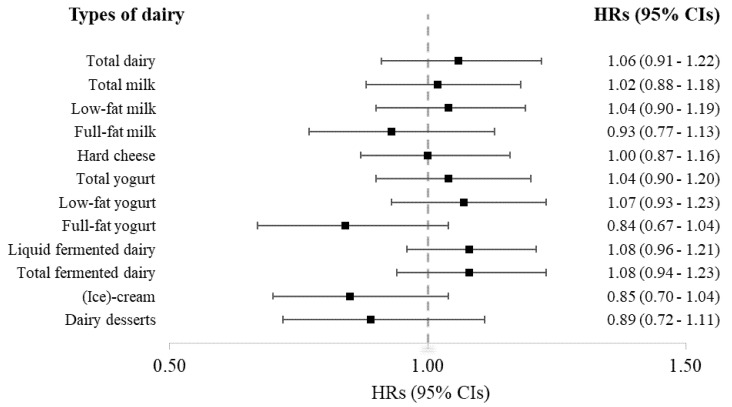
Hazard ratios (HRs) and 95% confidence intervals (95% CIs) for type 2 diabetes per 1-SD increment in (types of) dairy consumption (g/d) in 3401 patients from the Alpha Omega Cohort. Dairy intakes were adjusted for total energy intake. Multivariable models were adjusted for age (y), sex, total energy intake (kcal/d), smoking (3 categories), physical activity (3 categories), educational level (3 categories), alcohol intake (4 categories), family history of diabetes, BMI (kg/m^2^) and dietary intakes (g/d) of whole grains, refined grains, potatoes, fruit, vegetables, total red and processed meat, sugar-sweetened beverages, coffee, tea and other dairy products, except for total dairy (model 4).

**Table 1 nutrients-13-03146-t001:** Baseline characteristics of 3401 post-MI patients from the Alpha Omega Cohort, overall and across categories of energy-adjusted total milk intake.

	Total Population (*n* = 3401)	Energy-Adjusted Milk Intake (g/d)		
		<50 (*n* = 1303)	≥50–150 (*n* = 1075)	≥150 (*n* = 1023)
Milk intake, median (g/d)	87	11	102	244
Females, % (*n*)	20 (680)	17 (219)	22 (231)	23 (230)
Age (y)	68.9 ± 5.5	68.5 ± 5.4	69.0 ± 5.5	69.1 ± 5.6
Time since last MI (y)	3.5 (1.6–6.3)	3.4 (1.6–6.3)	3.8 (1.6–6.4)	3.5 (1.5–6.3)
Smoking, % (*n*)				
Never	17 (572)	16 (202)	17 (181)	19 (189)
Former	67 (2278)	68 (885)	67 (721)	66 (672)
Current	16 (550)	17 (216)	16 (172)	16 (162)
Physical activity, % (*n*) ^1^				
Low	39 (1314)	39 (501)	41 (433)	37 (380)
Intermediate	21 (723)	20 (264)	22 (236)	22 (223)
High	40 (1347)	41 (533)	38 (401)	41 (413)
Educational level, % (*n*)				
Low	56 (1888)	55 (714)	54 (577)	59 (597)
Intermediate	32 (1070)	32 (414)	34 (359)	29 (297)
High	13 (428)	13 (168)	13 (136)	12 (124)
BMI, (kg/m^2^)	27.4 ± 3.5	27.2 ± 3.5	27.5 ± 3.4	27.4 ± 3.5
Obesity, % (*n*) ^2^	20 (685)	19 (247)	21 (222)	21 (216)
Abdominal obesity, % (*n*) ^3^	52 (1767)	50 (645)	55 (586)	53 (536)
Alcohol consumption, % (*n*) ^4^				
Abstainers	4 (145)	4 (53)	4 (47)	4 (45)
Light	49 (1674)	46 (602)	48 (511)	55 (561)
Moderate	30 (1011)	30 (394)	32 (342)	27 (275)
Heavy	17 (571)	20 (254)	16 (175)	14 (142)
Systolic blood pressure, mmHg	141.7 ± 21.5	142.0 ± 21.9	142.0 ± 21.3	141.0 ± 21.2
Diastolic blood pressure, mmHg	80.7 ± 11.1	80.6 ± 11.3	81.2 ± 11.1	80.4 ± 11.0
Hypertension, % (*n*) ^5^	95 (3228)	95 (1239)	95 (1018)	95 (971)
Antihypertensive drugs, % (*n*)	89 (3024)	90 (1167)	89 (952)	89 (905)
Plasma glucose, mmol/L ^6^	5.59 ± 1.00	5.59 ± 1.03	5.60 ± 0.98	5.58 ± 0.96
Family history of T2D, % (*n*) ^7^	16 (559)	17 (218)	14 (150)	19 (191)
Serum blood lipids, mmol/L ^6^				
Total cholesterol ^8^	4.72 ± 0.95	4.71 ± 0.95	4.71 ± 0.94	4.74 ± 0.95
LDL cholesterol ^9^	2.59 ± 0.81	2.57 ± 0.80	2.59 ± 0.82	2.63 ± 0.83
HDL cholesterol ^8^	1.30 ± 0.34	1.31 ± 0.35	1.30 ± 0.33	1.30 ± 0.35
Triacylglycerol ^8^	1.84 ± 0.97	1.84 ± 0.98	1.86 ± 1.01	1.80 ± 0.91
Hypercholesterolemia, % (*n*) ^10^	96 (3271)	96 (1252)	97 (1037)	96 (982)
Statin use, % (*n*)	87 (2953)	87 (1136)	88 (944)	85 (873)

Values are means ± SD for normally distributed data, medians (IQRs) for skewed data or % (*n*) for categorical data. Abbreviations: BMI, body mass index; HDL, high density lipoprotein; LDL, low-density lipoprotein; METs, metabolic equivalent of task; MI, myocardial infarction; T2D, type 2 diabetes. Data missing for <1% of the cohort, except when stated otherwise. ^1^ Low: ≤3 MET, intermediate: >3 MET on 1–<5 d/wk, and high: >3 MET on ≥5 d/wk. ^2^ BMI ≥ 30 kg/m2. ^3^ Waist circumference ≥ 88 cm in women and ≥102 cm in men. ^4^ Light: >0–≤10 g/d for males or >0–≤5 g/d for females, moderate: >10–≤30 g/d for males or >5–≤15 g/d for females, and heavy: >30 g/d for males or >15 g/d for females. ^5^ Systolic blood pressure ≥ 140 mm/Hg or diastolic blood pressure ≥ 90 mm/Hg or use of antihypertensive drugs. ^6^ Fasting < 4 h (*n* = 1718), fasting 4 to <8 h (*n* = 339), fasting 8 to <12 h (*n* = 101), fasting ≥ 12 h (*n* = 1085), fasting status unknown (*n* = 20) or missing (*n* = 138). ^7^ One or both parents with diabetes. ^8^ Missing data for 76 patients. ^9^ Missing data for 198 patients. ^10^ Blood total cholesterol ≥ 5 mmol/L or use of lipid-lowering drugs.

**Table 2 nutrients-13-03146-t002:** HRs (95% CIs) for type 2 diabetes according to categories of types of dairy consumption in 3401 patients from the Alpha Omega Cohort ^1^.

	Categories of Dairy Consumption (g/d)			
	Low	Intermediate	High	Ρ for Trend ^2^
Total dairy	<200	≥200–<400	≥400	
Median intake, g/d	137	283	527	
Patients/events (*n*)	1004/48	1559/86	838/52	
Person-years	3159	4923	2632	
Model 1	1	1.15 (0.80–1.65)	1.31 (0.88–1.95)	0.17
Model 2	1	1.14 (0.79–1.64)	1.26 (0.84–1.88)	0.19
Model 3	1	1.18 (0.82–1.70)	1.32 (0.88–1.99)	0.18
Total milk	<50	≥50–<150	≥150	
Median intake, g/d	11	102	244	
Patients/events (*n*)	1303/65	1075/65	1023/56	
Person-years	4108	3366	3241	
Model 1	1	1.22 (0.86–1.72)	1.09 (0.76–1.56)	0.68
Model 2	1	1.19 (0.84–1.68)	1.07 (0.75–1.53)	0.75
Model 3	1	1.23 (0.87–1.74)	1.10 (0.76–1.58)	0.65
Model 4	1	1.23 (0.87–1.74)	1.10 (0.77–1.59)	0.62
Low-fat milk	<50	≥50–<150	≥150	
Median intake, g/d	10	104	249	
Patients/events (*n*)	1492/78	969/55	940/53	
Person-years	4693	3045	2976	
Model 1	1	1.08 (0.76–1.53)	1.07 (0.75–1.52)	0.60
Model 2	1	1.07 (0.75–1.51)	1.05 (0.74–1.49)	0.71
Model 3	1	1.09 (0.77–1.55)	1.08 (0.76–1.54)	0.55
Model 4	1	1.09 (0.77–1.55)	1.08 (0.76–1.54)	0.55
Hard cheese	<15	≥15–<30	≥30	
Median intake, g/d	9	21	48	
Patients/events (*n*)	1397/73	1174/71	830/42	
Person-years	4396	3694	2624	
Model 1	1	1.16 (0.83–1.63)	0.96 (0.66–1.41)	0.80
Model 2	1	1.14 (0.81–1.61)	0.92 (0.62–1.34)	0.61
Model 3	1	1.15 (0.82–1.62)	0.93 (0.63–1.36)	0.65
Model 4	1	1.15 (0.81–1.62)	0.92 (0.63–1.36)	0.64
Total yogurt	<25	≥25–<50	≥50	
Median intake, g/d	8	36	86	
Patients/events (*n*)	1271/70	635/31	1495/85	
Person-years	4008	1999	4707	
Model 1	1	0.88 (0.58–1.35)	1.04 (0.78–1.43)	0.71
Model 2	1	0.90 (0.58–1.38)	1.04 (0.75–1.45)	0.68
Model 3	1	0.91 (0.59–1.41)	1.07 (0.77–1.49)	0.58
Model 4	1	0.92 (0.60–1.41)	1.06 (0.76–1.48)	0.63
Low-fat yogurt	<25	≥25–<50	≥50	
Median intake, g/d	5	37	90	
Patients/events (*n*)	1555/81	473/22	1373/83	
Person-years	4895	1509	4310	
Model 1	1	0.88 (0.54–1.41)	1.17 (0.86–1.59)	0.27
Model 2	1	0.87 (0.54–1.41)	1.17 (0.85–1.60)	0.27
Model 3	1	0.90 (0.55–1.45)	1.20 (0.88–1.65)	0.21
Model 4	1	0.90 (0.55–1.45)	1.19 (0.86–1.64)	0.24
Total fermented dairy	<50	≥50–<100	≥100	
Median intake, g/d	26	73	185	
Patients/events (*n*)	570/33	631/33	2200/120	
Person-years	1807	1978	6930	
Model 1	1	0.90 (0.55–1.47)	0.94 (0.64–1.40)	0.92
Model 2	1	0.88 (0.54–1.44)	0.92 (0.62–1.37)	0.85
Model 3	1	0.89 (0.54–1.46)	0.94 (0.63–1.41)	0.95
Model 4	1	0.89 (0.54–1.47)	0.95 (0.63–1.42)	0.96
Dairy desserts	<30	≥30–<60	≥60	
Median intake, g/d	13	44	87	
Patients/events (*n*)	1003/56	1040/55	1358/75	
Person-years	3184	3276	4254	
Model 1	1	0.95 (0.64–1.39)	1.00 (0.70–1.44)	0.90
Model 2	1	0.94 (0.63–1.39)	1.02 (0.70–1.47)	0.83
Model 3	1	0.94 (0.63–1.38)	1.05 (0.73–1.53)	0.68
Model 4	1	0.94 (0.63–1.39)	1.06 (0.73–1.54)	0.66

^1^ Hazard ratios and 95% confidence intervals from Cox regression. Dairy intakes were adjusted for total energy intake. ^2^ Linear trends were assessed by treating the median value of each category of dairy as a continuous variable in the models. Model 1 was adjusted for age (y), sex and total energy intake (kcal/d). Model 2 was additionally adjusted for smoking (3 categories), physical activity (3 categories), educational level (3 categories), alcohol intake (4 categories), family history of diabetes and BMI (kg/m^2^). Model 3 was additionally adjusted for dietary intakes (g/d) of whole grains, refined grains, potatoes, fruit, vegetables, total red and processed meat, sugar-sweetened beverages, coffee and tea. Model 4 was additionally adjusted for intake of other dairy products (g/d), except for total dairy.

## Data Availability

The data presented in this study, code book, and analytic code will be made available upon request pending application and approval.

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
