# Peer review of "Dairy Consumption and 3-Year Risk of Type 2 Diabetes after Myocardial Infarction: A Prospective Analysis in the Alpha Omega Cohort"

_nutrients, 2021, doi:10.3390/nu13093146_

Round 1
Reviewer 1 Report
This is a well-written paper and I enjoyed reading it. Although a "negative" finding I think it is important. Good luck!
Author Response
Response to Reviewer 1 Comments
Point 1: This is a well-written paper and I enjoyed reading it. Although a "negative" finding I think it is important. Good luck!
Response 1: Thank you for carefully reading our manuscript and and for emphasizing the relevance also of negative findings.

Reviewer 2 Report
The manuscript from Maria Gorety Jacobo Cejudo reports on a population-based study carried out with the aim to verify the potential role for dairy consumption in the prevention of type 2 diabetes (T2D).
Data are potentially interesting. However, several points need to be better specified in order the manuscript can be improved:
1) The cohort of population enrolled for the analysis is 80% male. Please better define how this imbalance may interfere with conclusion for such a population-based study.
2) Please specify the international guidelines and criteria used for the definition of Diabetes
3) A better specification of the nutritional composition and daily consumption of different dairy is absolutely required in order to better understand the correlation with the development of DMT2.
Author Response
Response to Reviewer 2 Comments
Point 1: The cohort of population enrolled for the analysis is 80% male. Please better define how this imbalance may interfere with conclusion for such a population-based study.
Response 1: Thank you for the comment. A limitation of this study is indeed the lower proportion of women in the Alpha Omega Cohort (AOC) (20%, vs 80% men). No selection on basis of gender was made for the inclusion of participants in the Alpha Omega Cohort. However, due to the higher prevalence of coronary heart disease (primary outcome) in men, female participants were less represented in the cohort. Because of the smaller sample of women, results from sex-stratified subgroup analysis should be interpreted with caution. We elaborated more on the generalizability of the results in the Discussion (lines 366-371), as follows:
“Most patients in our study were Dutch men. Therefore, results cannot merely be translated to people of non-Caucasian origin and to women. Yet, although dairy consumption and T2D risk may vary by sex, we found no significant interaction terms between dairy and sex in our statistical analysis. Also, we have no biological explanation why men would respond differently to dairy intake than women.”
Point 2: Please specify the international guidelines and criteria used for the definition of Diabetes
Response 2: We clarified the criteria that we used for the definition of type 2 diabetes (lines:105-116), as follows:
“Diabetes at baseline was defined based on WHO guidelines [27]: plasma glucose levels ≥7.0 mmol/L (126 mg/dL) in fasting conditions (≥4 h), or ≥11.1 mmol/L (200 mg/dL) in non-fasting conditions, or having received the diagnosis from a physician (based on WHO guidelines), and/or taking antidiabetic drugs. New cases of T2D were ascertained based on a self-reported physician’s diagnosis and/or start of antidiabetic medication use. T2D was assessed by telephone interviews at 12 and 24 months, or during examinations at 20 months (in a subsample of 810 participants), or 40 months, with a median (interquartile range: IQR) follow-up of 40 (37.0-41.0) months. For the cases in which the date of diagnosis or start of medication was not available, it was calculated as the midpoint between two interview dates (10% of cases) [28]. Because incident T2D during early follow-up (i.e., initial trial phase) was the main outcome for this study, prevalent cases of diabetes at baseline were excluded.”
Point 3: A better specification of the nutritional composition and daily consumption of different dairy is absolutely required in order to better understand the correlation with the development of DMT2.
Response 3: The nutritional composition of dairy products is briefly described in the Discussion (lines 304-313), and we also elaborated more in the Introduction (lines 47-48). Furthermore, we added detailed information on the daily consumption of different dairy products in Supplemental Table S2 “Baseline dietary intake of 3401 post-MI patients from the Alpha Omega Cohort, overall and across categories of energy-adjusted total milk consumption”.

Round 2
Reviewer 2 Report
The manuscript has been improved in the revised version and now appears suitable of publication